# The Role of BiP and the IRE1α–XBP1 Axis in Rhabdomyosarcoma Pathology

**DOI:** 10.3390/cancers13194927

**Published:** 2021-09-30

**Authors:** Mahmoud Aghaei, Ahmad Nasimian, Marveh Rahmati, Philip Kawalec, Filip Machaj, Jakub Rosik, Bhavya Bhushan, S. Zahra Bathaie, Negar Azarpira, Marek J. Łos, Afshin Samali, David Perrin, Joseph W. Gordon, Saeid Ghavami

**Affiliations:** 1Department of Clinical Biochemistry, School of Pharmacy and Pharmaceutical Sciences, Isfahan University of Medical Sciences, Isfahan 81746-73461, Iran; parsa16657@yahoo.com; 2Department of Human Anatomy and Cell Science, University of Manitoba College of Medicine, Winnipeg, MB R3E 0V9, Canada; nnasimiann@gmail.com (A.N.); kawalecp@myumanitoba.ca (P.K.); machajf@edu.pum.edu.pl (F.M.); jakubrosik@edu.pum.edu.pl (J.R.); bhavya.bhushan@mail.mcgill.ca (B.B.); 3Department of Clinical Biochemistry, Faculty of Medical Sciences, Tarbiat Modares University, Tehran 14155-331, Iran; bathai_z@modares.ac.ir; 4Cancer Biology Research Center, Tehran University of Medical Sciences, Tehran 14197-33141, Iran; m_rahmati@tums.ac.ir; 5Institute for Natural Products and Medicinal Plants, Tarbiat Modares University, Tehran 14155-331, Iran; 6Transplant Research Center, Shiraz University of Medical Sciences, Shiraz 7134814336, Iran; negarazarpira@gmail.com; 7Biotechnology Center, Silesian University of Technology, 44-100 Gliwice, Poland; mjelos@gmail.com; 8Apoptosis Research Centre, School of Natural Sciences, National University of Ireland, H91 W2TY Galway, Ireland; afshin.samali@nuigalway.ie; 9Section of Orthopaedic Surgery, Department of Surgery, University of Manitoba, Winnipeg, MB R3E 0V9, Canada; david.perrin@umanitoba.ca; 10College of Nursing, Rady Faculty of Health Science, University of Manitoba, Winnipeg, MB R3E 0V9, Canada; joseph.gordon@umanitoba.ca; 11Research Institutes of Oncology and Hematology, Cancer Care Manitoba-University of Manitoba, Winnipeg, MB R3E 0V9, Canada; 12Biology of Breathing Theme, Children Hospital Research Institute of Manitoba, University of Manitoba, Winnipeg, MB R3E 0V9, Canada; 13Autophagy Research Center, Shiraz University of Medical Sciences, Shiraz 7134845794, Iran; 14Faculty of Medicine, Katowice School of Technology, 40-555 Katowice, Poland

**Keywords:** GRP78, IRE1, spliced XBP1, rhabdomyosarcoma, unfolded protein response

## Abstract

**Simple Summary:**

The expression of BiP (GRP78), spliced XBP1 (sXBP1, nuclear XBP1), and IRE1α, were significantly associated with Rhabdomyosarcoma (RMS) in the four main RMS subtypes: alveolar (ARMS), embryonal (ERMS), pleomorphic (PRMS) and sclerosing/spindle cell (SRMS) RMS; (*n* = 192) compared to normal skeletal muscle tissues (*n* = 16). There was a significant correlation between BiP expression and the lymph node score, and between IRE1α, cytosolic XBP1 and sXBP1 expression and the stage score in all of the types of RMS. BiP and sXBP1 expression were significantly associated with all of the subtypes of RMS, whereas IRE1α was associated with ARMS, PRMS and ERMS, and cytosolic XBP1 expression was associated with ARMS and SRMS. There were correlations between BiP expression and the lymph node score in ARMS, and sXBP1 expression and the tumor score in PRMS.

**Abstract:**

Background: Rhabdomyosarcoma (RMS) is the most common soft-tissue sarcoma in children, and is associated with a poor prognosis in patients presenting with recurrent or metastatic disease. The unfolded protein response (UPR) plays pivotal roles in tumor development and resistance to therapy, including RMS. Methods: In this study, we used immunohistochemistry and a tissue microarray (TMA) on human RMS and normal skeletal muscle to evaluate the expression of key UPR proteins (GRP78/BiP, IRE1α and cytosolic/nuclear XBP1 (spliced XBP1-sXBP1)) in the four main RMS subtypes: alveolar (ARMS), embryonal (ERMS), pleomorphic (PRMS) and sclerosing/spindle cell (SRMS) RMS. We also investigated the correlation of these proteins with the risk of RMS and several clinicopathological indices, such as lymph node involvement, distant metastasis, tumor stage and tumor scores. Results: Our results revealed that the expression of BiP, sXBP1, and IRE1α, but not cytosolic XBP1, are significantly associated with RMS (BiP and sXBP1 *p*-value = 0.0001, IRE1 *p*-value = 0.001) in all of the studied types of RMS tumors (*n* = 192) compared to normal skeletal muscle tissues (*n* = 16). In addition, significant correlations of BiP with the lymph node score (*p* = 0.05), and of IRE1α (*p* value = 0.004), cytosolic XBP1 (*p* = 0.001) and sXBP1 (*p* value = 0.001) with the stage score were observed. At the subtype level, BiP and sXBP1 expression were significantly associated with all subtypes of RMS, whereas IRE1α was associated with ARMS, PRMS and ERMS, and cytosolic XBP1 expression was associated with ARMS and SRMS. Importantly, the expression levels of IRE1α and sXBP1 were more pronounced in ARMS than in any of the other subtypes. The results also showed correlations of BiP with the lymph node score in ARMS (*p* value = 0.05), and of sXBP1 with the tumor score in PRMS (*p* value = 0.002). Conclusions: In summary, this study demonstrates that the overall UPR is upregulated and, more specifically, that the IRE1/sXBP1 axis is active in RMS. The subtype and stage-specific dependency on the UPR machinery in RMS may open new avenues for the development of novel targeted therapeutic strategies and the identification of specific tumor markers in this rare but deadly childhood and young-adult disease.

## 1. Introduction

Rhabdomyosarcoma (RMS) is a cancer of skeletal muscle tissue, and is the most common childhood soft tissue sarcoma; it originates from primitive mesenchymal cells. In children, it is the third most common extracranial solid neoplasm, accounting for 5% of pediatric malignancies, whereas it constitutes <1% of solid tumors in adults [1,2,3,4].

Macroscopically, RMS tumors grow as poorly circumscribed, infiltrative, white masses [5,6]. The 2013 WHO classification of soft tissue and bone tumors categorizes RMS into four subtypes: embryonal RMS (ERMS), alveolar RMS (ARMS), spindle cell/sclerosing RMS (SRMS), and pleomorphic RMS (PRMS) [7]. Even though these tumors all fall under RMS, the subtypes are driven by different molecular mechanisms, and present clinicians with distinct challenges.

The treatment of RMS presents unique challenges due to the rarity of the disease, its various anatomical sites, and its poor response to treatment in high risk cases. Children with high-risk RMS and recurrent disease have 5-year survival rates of less than 30% and 17%, respectively [8,9]. Due to these bleak outcomes, further targeted therapy strategies are under investigation. RMS is a neoplasm in which interfering with the unfolded protein response (UPR) could improve the treatment outcome and survival [10].

The UPR is a conserved cellular stress response mechanism [11] which is elicited by the accumulation of misfolded/unfolded proteins in the lumen of the endoplasmic reticulum (ER) [12].

During the UPR, three ER transmembrane proteins—namely activating transcription factor 6 (ATF6), inositol requiring enzyme 1 alpha (IRE1α), and protein kinase RNA-like ER kinase (PERK) [13,14]—are activated to orchestrate the cellular response to stress. IRE1, a type I transmembrane protein, possesses both RNase and kinase activities [15,16]. Activated IRE1α catalyzes the splicing of XBP1 mRNA [16,17], the biosynthesis of which is upregulated by ATF6 [17,18]. The spliced isoform (sXBP1) itself is a bZIP transcription factor involved in the UPR [16,19]. The ER sensors are deactivated via direct interaction with GRP78 (BiP) [20]. The dissociation of BiP from the stress sensors is thought to be responsible for their activation during the UPR, and the process is negatively regulated if the UPR increases its expression [21,22]. 

RMS cells exhibit competent UPR signaling [10,23], which might contribute to treatment failures and poor overall patient survival [24]. In the current study, we performed IHC on human tissue microarrays of different types of RMS—including ARMS, ERMS, SRMS, and PRMS—to investigate the involvement of the general UPR pathway and, more specifically, the IRE1-sXBP (sXBP refers to nuclear XBP axis [20,25]); we compared the results with findings in normal skeletal muscle. We demonstrate the importance of the UPR in RMS, and propose that interfering with the UPR could be of interest for future anticancer drug development. This study provides potential new therapeutic strategies for the treatment of RMS to improve the overall patient prognosis. 

## 2. Material and Methods

### 2.1. Tissue Microarray

The tissue microarray (TMA) analysis was performed on RMS and normal (control) striated muscle samples using a rhabdomyosarcoma tissue array kit (104 cases/208 cores) purchased from US Biomax, Inc. (cat. no. SO2082a; RKV, MD USA); information on tumors, nodes and metastases (TNM), clinical stages and pathology grades was provided. The TMA contained 15 SRMS, 27 ERMS, 30 PRMS, and 24 ARMS cases, and 8 normal skeletal muscle tissue cores (in duplicate). Thus, a total of 208 tissue cores were featured on a single slide, as two cores were obtained from each case/control. The histopathological characteristics of all of the samples are shown in Appendix A.

### 2.2. Immunohistochemistry

TMA slides (5 μm thick) were deparaffinized in 60 °C for 30 min and subsequently rehydrated in xylene with a gradient alcohol series. Heat-mediated antigen retrieval (0.01 M sodium citrate buffer, pH 6.0) was performed as described previously [22,26,27]. In order to eliminate background interference, the slides were washed with phosphate-buffered saline (PBS) and blocked with blocking solution (1.5 mL Maleic Acid Buffer, 0.5 mL FBS, 0.5 mL stock blocking solution, 50 μL 10% Tween-20, and 2.5 mL PBS) at room temperature for 30 min. After washing with PBS, the slides were incubated in freshly prepared 3% H_2_O_2_ to eliminate endogenous hydrogen peroxidases. The TMA slides were incubated with Avidin blocking solution (Vector SP-2001, 15 min) and then with Biotin blocking solution (Vector WP-2001, 15 min). Next, the slides were incubated overnight (4 °C) with mouse mAb against IRE1 (1:100; cat. no. ab96481; Abcam), rabbit mAb against BiP (1:200; cat. no. #3177, Cell Signaling; Danvers, MA USA), and rabbit polyclonal antibody against cytosolic and nuclear XBP1 (1:200; cat. no. ab37152; Abcam). Following thorough washing with PBS, the slides were incubated with biotinylated secondary antibody (corresponding to the primary antibody) for 1 h at room temperature, washed with PBS, and incubated with horseradish peroxidase-labeled streptavidin (1:200) for 30 min at room temperature. Finally, the slides were incubated with a 3,3′-diaminobenzidine-peroxidase substrate for 2 min at room temperature and counterstained with Mayer Hematoxylin (Vector H-3404; 10 drops in 1.25 mL PBS) for 1–4 min. In order to exclude any nonspecific staining of the secondary antibodies, negative controls were performed without the addition of any primary antibody.

### 2.3. Tissue Microarray Scoring

The immunohistochemistry (IHC) results were blindly evaluated by three independent pathologists who scored the samples based on the intensity of the staining [none (N), weak (W), moderate (M), and strong (S)]. sXBP1 was evaluated by assessing the nuclear staining of XBP1. 

### 2.4. Statistical Analysis

All of the data are expressed as n (%), and are compared using a χ^2^ test or Fisher’s exact test with SPSS software (version 16.0; SPSS, Inc., Chicago, IL, USA). All of the *p*-values are presented as two-tailed; *p* < 0.05 was considered to indicate a difference of statistical significance.

## 3. Results

### 3.1. UPR Related Proteins Are Associated with ARMS, ERMS, PRMS, and SRMS

We investigated and compared the expression of UPR-related proteins (BiP, IRE1α, and XBP1) in normal and various RMS-subtype skeletal muscle samples to identify a potential link between UPR and the pathogenesis of RMS. Our results revealed that the protein expression levels of BiP (*p*-value = 0.0001, Table 1, Figure 1A,B), IRE1 (*p* value = 0.001, Table 1, Figure 2A,B), and sXBP1 (*p* value = 0.0001, Table 1, Figure 3A,B) are associated with RMS; no correlation between cytosolic XBP1 and RMS could be observed when all of the subtypes were pooled together (*p* value = 0.41, Table 1, Figure 4A,B). Furthermore, significant correlations between BiP expression and the lymph node score (Table 1, LN0, LN1, *p* value 0.05), IRE1 expression and the stage score (Table 1, II, III, IV, *p* value 0.001), and cytosolic XBP1 and sXBP1 and the stage score (Table 1, *p* value = 0.0001 for both) were evident in the samples obtained from RMS patients. There were no significant associations between BiP expression and the stage score (*p* value = 0.245), the size of the tumor (*p* value = 0.21), and the distant metastasis score (*p* value =0.21) (Table 1). Although it was associated with the stage score, neither IRE1α nor cytosolic XBP1 expression levels were correlated with lymph node (*p* values 0.78 and 0.58), distant metastasis (*p* values 0.49 and 0.77) or tumor scores (*p* values 0.49 and 0.77) in RMS patients (Table 1). 

### 3.2. BiP Expression Is Associated with ARMS, ERMS, PRMS and SRMS

We evaluated whether BiP expression is associated with specific subtypes of RMS. Our results showed that BiP expression is increased in ARMS (Table 2; *p* value = 0.0001), PRMS (Table 2; *p* value = 0.0001), ERMS (Table 2; *p* value = 0.0001) and SRMS (Table 2; *p* value = 0.0001). Furthermore, we observed a significant association of BiP expression with the lymph node score (*p* value = 0.05), but not with distant metastasis (*p* value = 0.24), tumor (*p* value = 0.86) or stage (*p* value = 0.14) scores in ARMS (Table 2). There were no significant associations between these parameters and BiP expression in PRMS, ERMS and SRMS samples (Table 2).

### 3.3. IRE1α Expression Is Associated with ARMS, ERMS and PRMS, but Not with SRMS

Our investigation showed that the expression of IRE1α was associated with ARMS (Table 3; *p* value = 0.0001), PRMS (Table 3; *p* value = 0.005) and ERMS (Table 3; *p* value = 0.001), but it is not associated with SRMS (Table 3, *p* value = 0.56). No correlations between IRE1α expression and the investigated clinicopathological factors (lymph node, distant metastasis, stage and tumor scores) could be observed in any of the RMS subtypes (Table 3).

### 3.4. Cytosolic XBP1 Expression Is Associated with ARMS and SRMS, but Not with ERMS and PRMS

Our analysis showed that cytosolic XBP1 expression was associated with ARMS (Table 4; *p* value = 0.034) and SRMS (Table 4; *p* value = 0.01), but not with ERMS (Table 4; *p* value = 0.88) or PRMS (Table 4; *p* value = 0.81). There were no associations between cytosolic XBP1 expression and the clinicopathological parameters in any of the RMS subtypes (Table 4).

### 3.5. sXBP1 Expression Is Associated with ARMS, ERMS, PRMS and SRMS

Finally, our results demonstrate that sXBP1 expression is associated with ARMS (Table 5, *p* value = 0.0001), PRMS (Table 5, *p* value = 0.003), ERMS (Table 5, *p* value = 0.009), and SRMS (Table 5, *p* value = 0.01). In addition, sXBP1 was correlated with the distant metastasis score in ARMS (*p* value = 0.04) and tumor score in PRMS (*p* value = 0.002); no significant associations between sXBP1 and the other clinicopathological factors were evident (Table 5).

## 4. Discussion

Although our understanding of the underlying mechanisms of and treatment approaches for RMS is gradually improving, the poor prognosis and failure of current therapies remain a medical challenge [28]. Here, for the first time, we provide evidence that the expression levels of BiP and the IRE1α/sXBP1 arm of the UPR are associated with different types of RMS. These findings are of clinical importance, as these proteins could represent potential therapeutic targets and/or prognosis markers for distinct RMS forms.

Our results revealed statistically significant associations of BiP, IRE1α, and sXBP1 with RMS. In addition, we demonstrated that the increase in the expression of sXBP1 is higher than that of cytoplasmic XBP1 in RMS tumors, which may reflect the functional activity of the IRE1α arm of the UPR in these tumors. Upon its activation, IRE1α cleaves XBP1 to render sXBP1, which in turn translocates to the nucleus to function as a transcription factor. Increased sXBP1 was strongly correlated with the poor survival of breast cancer patients, whereas cytosolic XBP1 expression had no relationship with patient survival [29].

The upregulation of BiP expression, as a master regulator of the UPR and the IRE1-XBP1 axis, strongly suggests that UPR is a hallmark of RMS tumors, which is in agreement with reports on the expression of UPR markers in several other cancers [30]. 

Regarding subtype analyses, we found that only BiP and sXBP1 are associated with all of the RMS subtypes (ARMS, PRMS, ERMS and SRMS), while IRE1α expression is associated with ARMS, PRMS and ERMS, and cytosolic XBP1 expression is associated with ARMS and SRMS. Therefore, the overall activation of UPR, as evidenced by an increase in the UPR master regulator BiP in our study, is a hallmark of all RMS subtypes, whereas dynamics in the expression of the IRE1-XBP1 arm are highly dependent on the RMS subtype. It should be noted that the increased expression levels of IRE1α and its substrate XBP1 (both nuclear and cytoplasmic) were more pronounced in ARMS than in any other RMS subtype. Importantly, these results are consistent with the findings of McCarthy et al., who reported that ARMS was highly sensitive to IRE1α inhibition compared to ERMS, which is more sensitive to PERK inhibition, suggesting the differential involvement of UPR arms in distinct subtypes of RMS [24]. Combined with our current observations, these findings could be of relevance for the design of targeted therapeutic strategies. 

In recent years, several studies have been focused on the connection of UPR pathways with clinicopathological factors and the onset and progression of cancers. For example, the upregulation of BiP has been associated with several clinicopathological factors, including drug resistance, angiogenesis and metastasis, a greater risk of cancer recurrence, and an overall decrease in patient survival [31]. In the current study, we also found a positive correlation between the upregulation of IRE1α and cytosolic/sXBP1 with the tumor stage in RMS; thus, concomitant with the progress in tumor stages, enhanced expression levels of these UPR markers were observed in all of the RMS samples. Although we did not find a significant relationship between metastasis and UPR markers for all of the RMS samples, subtype analysis revealed that sXBP1 was associated with metastasis in ARMS, the most aggressive form of RMS, which is also very lethal and less susceptible to therapeutic success compared to the other subtypes [32]. These results suggest that the IRE1α arm of UPR is more profoundly associated with aggressiveness and the progression of the disease, which further confirms that the targeting of IRE1α could be considered for the treatment of RMS, especially ARMS. This is in line with other studies that confirmed the role of UPR in tumor progression and metastasis [33]. 

The correlation between the IRE1α/sXBP1 arm of the UPR and BiP and lymph node involvement is described in a few studies. The downregulation of sXBP1 is significantly correlated with lymph node metastasis in papillary thyroid cancer [34]. The inhibition of BiP upregulation led to the suppression of tumor cell growth, invasion and metastasis in a xenograft mouse model [35], suggesting the relevance of BiP upregulation in lymph node metastasis. Despite these reports, our results revealed that IRE1α, cytosolic XBP1 and sXBP1 have no correlation with the lymph node score in RMS tumors. We found that only GRP78/BiP can be significantly associated with the lymph node score in RMS and some subtypes of ARMS.

The relationship between BiP, IRE1-XBP1 and tumor size in cancers might not be clear-cut, and likely depends on the marker and type of cancer studied. For instance, sXBP1 has no correlation with tumor size in breast cancer [36], while the increased expression of BiP and GRP94 was correlated with a larger tumor size and enhanced metastatic capability in esophageal adenocarcinomas [37]. In our study, a positive correlation between tumor size and sXBP1 was observed in PRMS samples. This further supports the supposition that UPR may have different roles depending on the subtype of the RMS, and indicates that a larger tumor size is not necessarily associated with higher UPR activity in all RMS tumors. 

## 5. Conclusions

In summary, the BiP, IRE1, and sXBP1 expression levels were significantly increased in skeletal muscle samples obtained from RMS patients compared to the control muscle samples. We demonstrated a strong correlation between BiP and all of the RMS subtypes. The most pronounced correlations were observed for IRE1α and sXBP1 with ARMS, highlighting the pathological and therapeutic significance of the IRE1α arm of UPR in this subtype. Clinicopathological analyses revealed that the IRE1α arm (IRE1 and cytosolic/sXBP1) correlated with tumor staging, and BiP correlated with lymph node involvement in all of the RMS samples. At the subtype level, only sXBP1 was associated with metastasis in ARMS and tumor size in PRMS; the other UPR markers did not show any correlation with these clinicopathological factors. The correlation of the RMS stage and subtype with specific elements of UPR extends our knowledge on disease mechanisms, identifies new diagnostic markers, and reveals potential therapeutic targets for this uncured disease. Based on our findings, the inhibition of IRE1α seems to be the most promising approach in ARMS, whereas others indicate that the inhibition of PERK could be a strategy of interest in ERMS [24]. The genetic knockdown of proteins participating in the UPR and/or utilizing inhibitors of these proteins (e.g., MKC8866 (IRE1α inhibitor) [38,39,40,41], AMGEN44, GSK-2606414 (PERK inhibitors) [25,39,42], and salubrinal (eIF2α inhibitor) [25]) in a mouse or zebrafish model of RMS will determine the effectiveness of UPR inhibition in RMS treatment, and will provide proper preliminary pre-clinical results; such UPR intervention studies will be part of our future research.

## Figures and Tables

**Figure 1 cancers-13-04927-f001:**
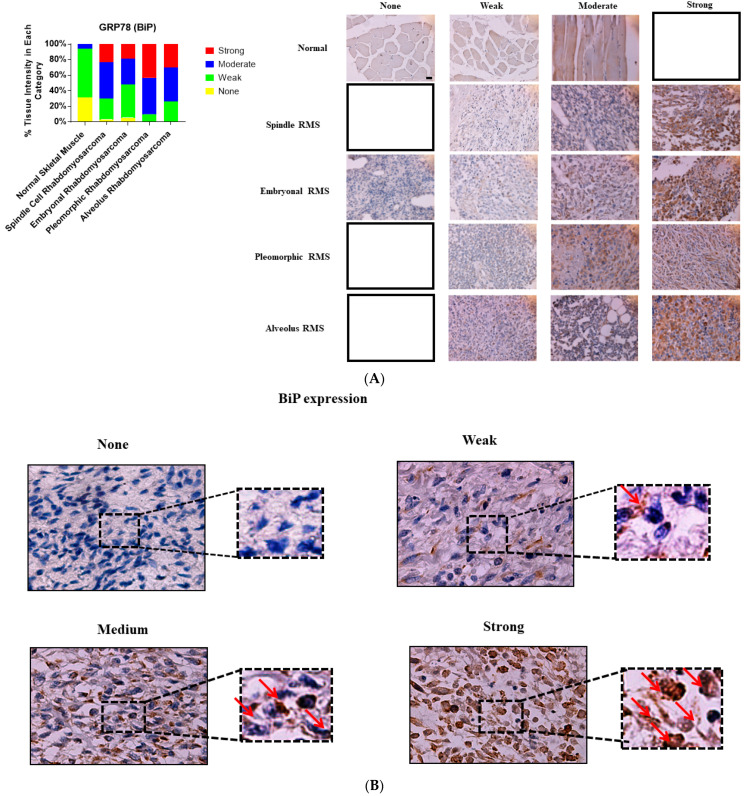
Immunohistochemical staining for BiP in human skeletal muscle and RMS tissue. (**A**,**B**). We detected BiP expression (red arrow) in a human RMS TMA; details on the TMA were outlined in the Section 2. Three independent pathologists blindly evaluated the immunohistochemical BiP expression (None = no staining; W: Weak staining; M: Moderate staining; S: Strong staining); the empty slots indicate that there were no samples in that specific group (None, W, M, or S). Our results show that BiP is associated with RMS (Table 1). The results also show that BiP expression is associated with ARMS, ERMS, PRMS, and SRMS (Table 2). The scale bar refers to 50 µM.

**Figure 2 cancers-13-04927-f002:**
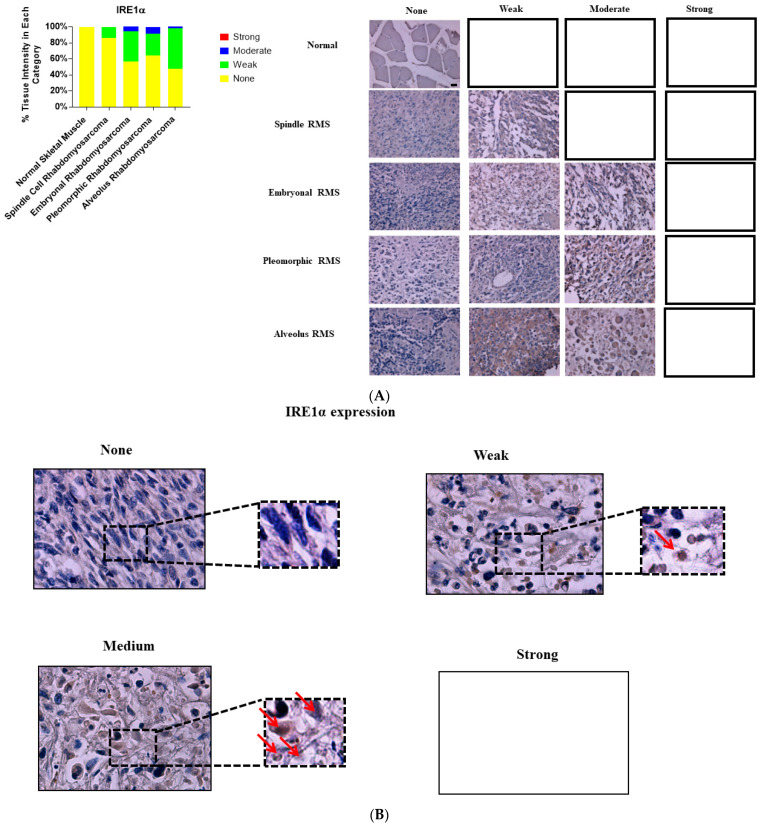
Immunohistochemical staining for IRE1α in human skeletal muscle and RMS tissue. (**A**,**B**). We detected IRE1α expression (red arrows) in a human RMS TMA; details on the TMA were mentioned in the Section 2. Three independent pathologists blindly evaluated the immunohistochemical IRE1α expression (None = no staining; W: Weak staining; M: Moderate staining; S: Strong staining); the empty slots indicate that no samples met the characteristics of that specific group. Our results show that IRE1α is associated with RMS (Table 1). The results also show that IRE1α expression is associated with ARMS, ERMS, PRMS, SRMS (Table 3). The scale bar refers to 50 µM.

**Figure 3 cancers-13-04927-f003:**
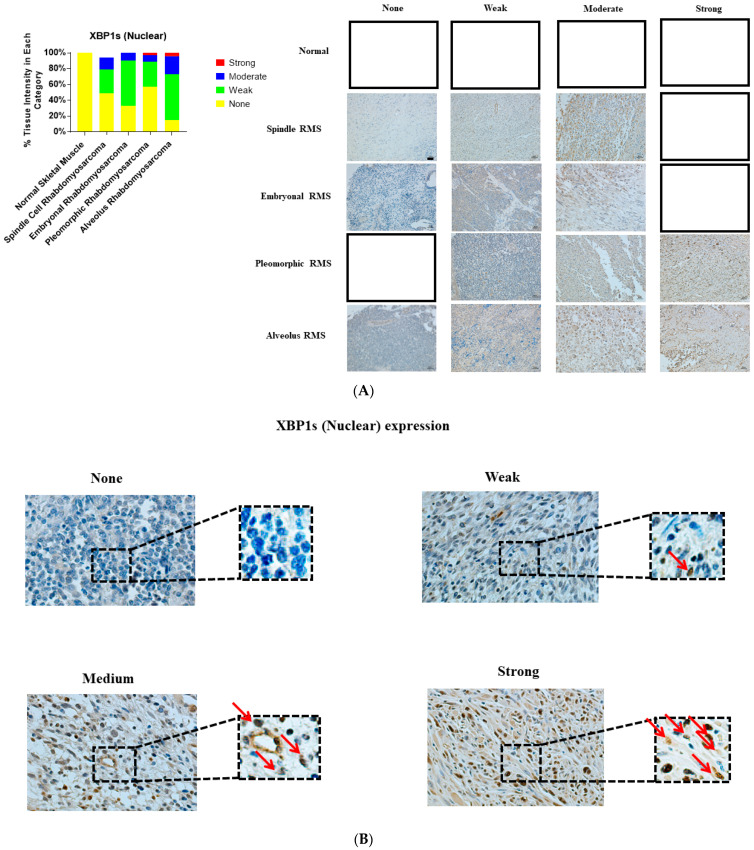
Immunohistochemical staining for sXBP1 in human skeletal muscle and RMS tissue. (**A**,**B**). We detected sXBP1 expression (red arrows) in human RMS TMA. Detailed information on the TMA was given in the Section 2. Three independent pathologists blindly evaluated the immunohistochemical sXBP1 expression (None = no staining; W: Weak staining; M: Moderate staining; S: Strong staining); the empty slots indicate that no samples matched the specifics of that group. Our results show that sXBP1 expression is associated with ARMS, ERMS, PRMS, and SRMS (Table 5). The scale bar refers to 50 µM.

**Figure 4 cancers-13-04927-f004:**
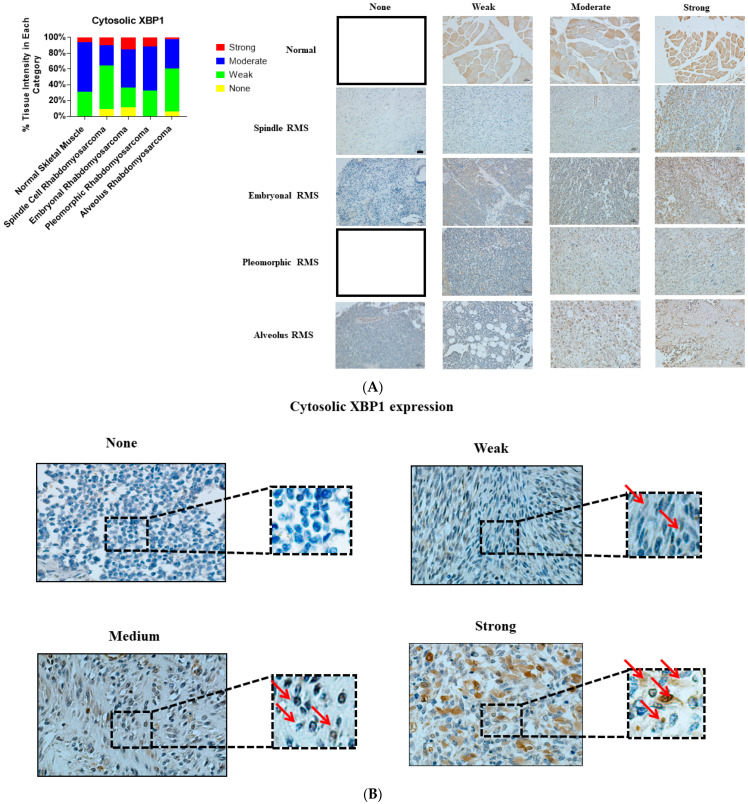
Immunohistochemical staining for cytosolic XBP1 in human skeletal muscle and RMS tissue. (**A**,**B**). We detected cytosolic XBP1 expression (red arrows) in a human RMS TMA. Details on the TMA were outlined in the Section 2. Three independent pathologists blindly evaluated the immunohistochemical cytosolic XBP1 expression (None = no staining; W: Weak staining; M: Moderate staining; S: Strong staining); the empty slots indicate that no samples matched the specifics of that group. Our results show that cytosolic XBP1 is associated with ARMS and SRMS (Table 4). The scale bar refers to 50 µM.

**Table 1 cancers-13-04927-t001:** Correlation of BiP, IRE1α, Cytosolic and XBP1s protein expression with the clinicopathological features of RMS.

Parameter	RMS (All Subtypes)	Normal Muscle	Lymph Node Score		Distant Metastasis Score		Stage Score			Tumor Score	
**BiP**											
	**RMS (all subtypes)**	**Normal**	**LN0**	**LN1**	**M0**	**M1**	**II**	**III**	**IV**	**T1a** **–** **T1b**	**T2a** **–** **T2b**
**N**	192	16	188	4	188	4	30	156	6	74	118
**% None**	2.1	31.2	2.1	0	2.1	0	3.3	1.9	0	2.7	1.7
**% W**	25	62.5	25.5	0	25.5	0	26.7	25.6	0	27.1	23.7
**% M**	42.7	6.3	43.1	25	42.6	50	46.7	41.7	50	40.5	44.1
**% S**	30.2	0	29.3	75	29.8	50	23.3	30.8	50	29.7	30.5
** *p* ** **-value**	0.0001		0.05		0.21		0.245			0.21	
**IRE1α**											
	**RMS (all subtypes)**	**Normal**	**LN0**	**LN1**	**M0**	**M1**	**II**	**III**	**IV**	**T1a** **–** **T1b**	**T2a** **–** **T2b**
**N**	192	16	188	4	188	4	30	156	6	74	118
**% None**	59.2	100	58.5	75	58.5	75	80	55.8	0	51.3	63.5
**% W**	36.1	0	36.7	25	36.7	25	20	38.5	0	41.9	33.1
**% M**	4.7	0	4.8	0	4.8	0	0	5.8	50	6.8	3.4
**% S**	0	0	0	0	0	0	0	0	50	0	0
** *p* ** **-value**	0.001		0.78		0.49		0.001			0.49	
**Cytosolic XBP1**											
	**RMS (all subtypes)**	**Normal**	**LN0**	**LN1**	**M0**	**M1**	**II**	**III**	**IV**	**T1a** **–** **T1b**	**T2a** **–** **T2b**
**N**	192	16	188	4	188	4	30	156	6	74	118
**% None**	6.3	0	6.4	0	6.4	0	53.3	5.8	0	8.2	5.1
**% W**	38.5	31.2	37.8	75	38.3	50	33.3	35.3	0	37.8	39
**% M**	44.3	62.5	45.2	0	44.1	50	13.4	48.1	50	37.8	48.3
**% S**	10.9	6.3	10.6	25	11.2	0	0	10.9	50	16.2	7.6
** *p* ** **-value**	0.41		0.58		0.77		0.0001			0.77	
**sXBP1**											
	**RMS (all subtypes)**	**Normal**	**LN0**	**LN1**	**M0**	**M1**	**II**	**III**	**IV**	**T1a** **–** **T1b**	**T2a** **–** **T2b**
**N**	192	16	188	4	188	4	30	156	6	74	118
**% None**	40.6	100	40.4	50	40.4	50	53.3	38.5	0	31.1	46.6
**% W**	44.8	0	45.8	0	44.7	50	33.3	47.4	0	51.3	40.7
**% M**	12.5	0	11.7	50	12.8	0	13.4	11.5	50	12.2	12.7
**% S**	2.1	0	2.1	0	2.1	0	0	2.6	50	5.4	0
** *p* ** **-Value**	0.0001		0.64		0.52		0.0001			0.52	

**Table 2 cancers-13-04927-t002:** Correlation of BiP protein expression with the clinicopathological features of RMS subtypes.

Parameter	RMS Subtype	Normal Muscle	Lymph Node Score		Distant Metastasis Score		Stage Score			Tumor Score	
**ARMS**											
	**ARMS**	**Normal**	**LN0**	**LN1**	**M0**	**M1**	**II**	**III**	**IV**	**T1a-T1b**	**T2a-T2b**
**N**	48	16	44	4	44	4	0	42	6	18	30
**% None**	0	31.2	0	0	0	0	0	0	0	0	0
**% W**	22.9	62.5	25	0	25	50	0	26.2	0	27.8	20
**% M**	45.8	6.3	47.7	25	45.5	50	0	45.2	50	38.9	50
**% S**	31.3	0	27.3	75	29.5	0	0	28.6	50	33.3	30
** *p* ** **-value**	0.0001		0.05		0.24		0.14			0.86	
**PRMS**											
	**PRMS**	**Normal**	**LN0**	**LN1**	**M0**	**M1**	**II**	**III**	**IV**	**T1a-T1b**	**T2a-T2b**
**N**	60	16	60	0	60	0	0	60	0	18	42
**% None**	0	31.2	0	0	0	0	0	0	0	0	0
**% W**	10	62.5	10	0	10	0	0	10	0	11.1	9.5
**% M**	46.7	6.3	46.7	0	46.7	0	0	46.7	0	27.8	54.8
**% S**	43.3	0	43.3	0	43.3	0	0	43.3	0	61.1	35.7
** *p* ** **-value**	0.0001		NA		NA		NA			0.09	
**ERMS**											
	**ERMS**	**Normal**	**LN0**	**LN1**	**M0**	**M1**	**II**	**III**	**IV**	**T1a-T1b**	**T2a-T2b**
**N**	54	16	52	0	52	0	0	50	2	26	26
**% None**	5.6	31.2	5.8	0	5.8	0	0	6	0	7.7	3.8
**% W**	42.6	62.5	42.3	0	42.3	0	0	44	0	38.5	46.2
**% M**	33.3	6.3	32.7	0	32.7	0	0	30	100	46.1	19.2
**% S**	18.5	0	19.2	0	19.2	0	0	20	0	7.7	30.8
** *p* ** **-value**	0.0001		NA		NA		0.56			0.33	
**SRMS**											
	**SRMS**	**Normal**	**LN0**	**LN1**	**M0**	**M1**	**II**	**III**	**IV**	**T1a-T1b**	**T2a-T2b**
**N**	30	16	30	0	30	0	30	0	0	10	20
**% None**	3.3	31.2	3.3	0	3.3	0	3.3	0	0	0	5
**% W**	26.7	62.5	26.7	0	26.7	0	26.7	0	0	20	30
**% M**	46.7	6.3	46.7	0	46.7	0	46.7	0	0	50	45
**% S**	23.3	0	23.3	0	23.3	0	23.3	0	0	30	20
** *p* ** **-value**	0.0001		NA		NA		NA			0.36	

**Table 3 cancers-13-04927-t003:** Correlation of IRE1α protein expression with the clinicopathological features of RMS subtypes.

Parameter	RMS Subtype	Normal Muscle	Lymph Node Score		Distant Metastasis Score		Stage Score			Tumor Score	
**ARMS**											
	**ARMS**	**Normal**	**LN0**	**LN1**	**M0**	**M1**	**II**	**III**	**IV**	**T1a-T1b**	**T2a-T2b**
**N**	48	16	44	4	44	4	0	42	6	18	30
**% None**	45.8	100	45.4	50	43.2	75	0	45.2	50	44.4	46.7
**% W**	52.1	0	52.3	50	54.5	25	0	52.4	50	55.6	50
**% M**	2.1	0	2.3	0	2.3	0	0	2.4	0	0	3.3
**% S**	0	0	0	0	0	0	0	0	0	0	0
** *p* ** **-value**	0.0001		0.83		0.27		0.79			0.96	
**PRMS**											
	**PRMS**	**Normal**	**LN0**	**LN1**	**M0**	**M1**	**II**	**III**	**IV**	**T1a-T1b**	**T2a-T2b**
**N**	60	16	60	0	60	0	0	60	0	18	42
**% None**	63.3	100	63.3	0	63.3	0	0	63.3	0	44.4	71.4
**% W**	28.3	0	28.3	0	28.3	0	0	28.3	0	44.4	21.4
**% M**	8.4	0	8.4	0	8.4	0	0	8.4	0	11.2	7.2
**% S**	0	0	0	0	0	0	0	0	0	0	0
** *p* ** **-value**	0.005		NA		NA		NA			0.09	
**ERMS**											
	**ERMS**	**Normal**	**LN0**	**LN1**	**M0**	**M1**	**II**	**III**	**IV**	**T1a-T1b**	**T2a-T2b**
**N**	54	16	52	0	52	0	0	50	2	26	26
**% None**	53.7	100	51.9	0	51.9	0	0	54	0	53.9	50
**% W**	40.7	0	42.3	0	42.3	0	0	40	100	34.6	50
**% M**	5.6	0	5.8	0	5.8	0	0	6	0	11.5	0
**% S**	0	0	0	0	0	0	0	0	0	0	0
** *p* ** **-value**	0.001		NA		NA		0.27			0.65	
**SRMS**											
	**SRMS**	**Normal**	**LN0**	**LN1**	**M0**	**M1**	**II**	**III**	**IV**	**T1a-T1b**	**T2a-T2b**
**N**	30	16	30	0	30	0	30	0	0	10	20
**% None**	83.3	100	80	0	80	0	80	0	0	70	85
**% W**	16.7	0	20	0	20	0	20	0	0	30	15
**% M**	0	0	0	0	0	0	0	0	0	0	0
**% S**	0	0	0	0	0	0	0	0	0	0	0
** *p* ** **-value**	0.56		NA		NA		NA			0.35	

**Table 4 cancers-13-04927-t004:** Correlation of Cytosolic XBP1 protein expression with the clinicopathological features of RMS subtypes.

Parameter	RMS Subtype	Normal Muscle	Lymph Node Score		Distant Metastasis Score		Stage Score			Tumor Score	
**ARMS**											
	**ARMS**	**Normal**	**LN0**	**LN1**	**M0**	**M1**	**II**	**III**	**IV**	**T1a-T1b**	**T2a-T2b**
**N**	48	16	44	4	44	4	0	42	6	18	30
**% None**	6.3	0	6.8	0	6.8	0	0	7.1	0	0	10
**% W**	54.2	31.2	52.3	75	54.5	50	0	54.8	50	61.1	50
**% M**	37.5	62.5	40.9	0	36.4	50	0	38.1	33.3	33.3	40
**% S**	2.1	6.3	0	25	2.3	0	0	0	16.7	5.6	0
** *p* ** **-value**	0.034		0.96		0.61		0.34			0.96	
**PRMS**											
	**PRMS**	**Normal**	**LN0**	**LN1**	**M0**	**M1**	**II**	**III**	**IV**	**T1a-T1b**	**T2a-T2b**
**N**	60	16	60	0	60	0	0	60	0	18	42
**% None**	0	0	0	0	0	0	0	0	0	0	0
**% W**	31.7	31.2	31.7	0	31.7	0	0	31.7	0	22.2	35.7
**% M**	56.7	62.5	56.7	0	56.7	0	0	56.7	0	55.6	57.1
**% S**	11.6	6.3	11.6	0	11.6	0	0	11.6	0	22.2	7.2
** *p* ** **-value**	0.81		NA		NA		NA			0.12	
**ERMS**											
	**ERMS**	**Normal**	**LN0**	**LN1**	**M0**	**M1**	**II**	**III**	**IV**	**T1a-T1b**	**T2a-T2b**
**N**	54	16	52	0	52	0	0	50	2	26	26
**% None**	11.1	0	11.6	0	11.6	0	0	12	0	11.5	11.5
**% W**	24.1	31.2	25	0	25	0	0	22	100	30.8	19.3
**% M**	46.3	62.5	44.2	0	44.2	0	0	46	0	34.6	53.8
**% S**	18.5	6.3	19.2	0	19.2	0	0	20	0	23.1	15.4
** *p* ** **-value**	0.88		NA		NA		0.26			0.88	
**SRMS**											
	**SRMS**	**Normal**	**LN0**	**LN1**	**M0**	**M1**	**II**	**III**	**IV**	**T1a-T1b**	**T2a-T2b**
**N**	30	16	30	0	30	0	30	0	0	10	20
**% None**	10	0	10	0	10	0	10	0	0	20	5
**% W**	53.3	31.2	53.3	0	53.3	0	53.3	0	0	40	60
**% M**	26.7	62.5	26.7	0	26.7	0	26.7	0	0	10	35
**% S**	10	6.3	10	0	10	0	10	0	0	30	0
** *p* ** **-value**	0.01		NA		NA		NA			0.79	

**Table 5 cancers-13-04927-t005:** Correlation of sXBP1 protein expression with the clinicopathological features of RMS subtypes.

Parameter	RMS Subtype	Normal Muscle	Lymph Node Score		Distant Metastasis Score		Stage Score			Tumor Score	
**ARMS**											
	**ARMS**	**Normal**	**LN0**	**LN1**	**M0**	**M1**	**II**	**III**	**IV**	**T1a-T1b**	**T2a-T2b**
**N**	48	16	44	4	44	4	0	42	6	18	30
**% None**	14.6	100	11.4	50	11.4	50	0	11.9	33.3	0	23.3
**% W**	58.3	0	63.6	0	59.1	50	0	61.9	33.3	66.7	53.4
**% M**	22.9	0	20.5	50	25	0	0	21.4	33.4	22.2	23.3s
**% S**	4.2	0	4.5	0	4.5	0	0	4.8	0	11.1	
** *p* ** **-value**	0.0001		0.74		0.04		0.62			0.06	
**PRMS**											
	**PRMS**	**Normal**	**LN0**	**LN1**	**M0**	**M1**	**II**	**III**	**IV**	**T1a-T1b**	**T2a-T2b**
**N**	60	16	60	0	60	0	0	60	0	18	42
**% None**	60	100	60	0	60	0	0	60	0	33.4	71.4
**% W**	31.7	0	31.7	0	31.7	0	0	31.7	0	44.4	26.2
**% M**	5	0	5	0	5	0	0	5	0	11.1	2.4
**% S**	3.3	0	3.3	0	3.3	0	0	3.3	0	11.1	0
** *p* ** **-value**	0.003		NA		NA		NA			0.002	
**ERMS**											
	**ERMS**	**Normal**	**LN0**	**LN1**	**M0**	**M1**	**II**	**III**	**IV**	**T1a-T1b**	**T2a-T2b**
**N**	54	16	52	0	52	0	0	50	2	26	26
**% None**	35.2	100	36.6	0	36.6	0	0	34	100	42.3	30.8
**% W**	53.7	0	51.9	0	51.9	0	0	54	0	50	53.8
**% M**	11.1	0	11.5	0	11.5	0	0	12	0	7.7	15.4
**% S**	0	0	0	0	0	0	0	0	0	0	0
** *p* ** **-value**	0.009		NA		NA		0.09			0.29	
**SRMS**											
	**SRMS**	**Normal**	**LN0**	**LN1**	**M0**	**M1**	**II**	**III**	**IV**	**T1a-T1b**	**T2a-T2b**
**N**	30	16	30	0	30	0	30	0	0	10	20
**% None**	53.4	100	53.4	0	53.4	0	53.4	0	0	50	55
**% W**	33.3	0	33.3	0	33.3	0	33.3	0	0	30	35
**% M**	13.3	0	13.3	0	13.3	0	13.3	0	0	20	10
**% S**	0	0	0	0	0	0	0	0	0	0	0
** *p* ** **-value**	0.01		NA		NA		NA			0.66	

## Data Availability

The data supporting the findings of this study are available in the Appendix A. To ensure transparency and allow readers to directly review the original immunohistochemistry data, all raw images, including those presented in Figure 1, Figure 2, Figure 3 and Figure 4, have been deposited in the Dryad digital repository. These can be accessed via the following DOI: https://datadryad.org/dataset/doi:10.5061/dryad.d2547d8c6.

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
