# Peer review of "The Role of BiP and the IRE1α–XBP1 Axis in Rhabdomyosarcoma Pathology"

_cancers, 2021, doi:10.3390/cancers13194927_

Round 1

Reviewer 1 Report

Major comments:

The authors use "nuclear" and "spliced" XBP1 interchangeably.  I am not familiar with data demonstrating that all nuclear XBP1 is spliced and all cytosolic XBP1 is unspliced.  If it exists, it should be referenced; otherwise, using the terms "nuclear" and "cytosolic" is preferred. 

Table 1 appears to have copied/pasted values for cytosolic & nuclear XBP1 when broken down by stage.

Why is there a discrepancy between Stage IV and metastasis?  Those should be identical.  

Minor comments:

Page 3, lines 82-84: the sentence needs to be reworded for grammatical correction.

Figure 1: typographical error, "alveolus" instead of "alveolar." This is repeated in all IHC panels.

Table 2: replace "unstatistical" with "NA" to maintain consistency.

Finally, I fear that while the discussion is very cogent, the conclusions regarding the use of inhibitors to change therapy are broadly overstated. 

Author Response

Question 1: The authors use "nuclear" and "spliced" XBP1 interchangeably.  I am not familiar with data demonstrating that all nuclear XBP1 is spliced and all cytosolic XBP1 is unspliced.  If it exists, it should be referenced; otherwise, using the terms "nuclear" and "cytosolic" is preferred.

Answer: The authors appreciate the constructive comment. As sXBP1 is widely used in place of “nuclear XBP1), we have changed “nuclear XBP1” to sXBP1 in the whole manuscript (we gave reference in the first place). We did not change it in the abstract as we were not able to provide citation in the abstract.

Question 2: Table 1 appears to have copied/pasted values for cytosolic & nuclear XBP1 when broken down by stage.

Answer: We appreciate the respected reviewer careful comment. We have checked the table 1 and corrected some numbers, which was wrong in the primary submission. We also checked all analysis in the revised version.

Question 3: Why is there a discrepancy between Stage IV and metastasis?  Those should be identical. 

Answer: We appreciate the respected reviewer careful comment. We have corrected the numbers in all tables. There was a some typo error while the tables were converted to the journal format. Actually, the number of patients with “Stage IV” was 6 based on the company information (supplementary table 1). 4 of them had metastasis but for 2 of them the metastasis was not reported.

Minor comments:

Question 4: Page 3, lines 82-84: the sentence needs to be reworded for grammatical correction.

Answer: We corrected lines 82-84 in the revised version.

Question 5: Figure 1: typographical error, "alveolus" instead of "alveolar." This is repeated in all IHC panels.

Answer: Figure 1 was corrected in the revised version.

Question 6: Table 2: replace "unstatistical" with "NA" to maintain consistency.

Answer: Table 2 was corrected based on the respected reviewer comment.

Question 7: Finally, I fear that while the discussion is very cogent, the conclusions regarding the use of inhibitors to change therapy are broadly overstated.

Answer: We totally agree with the reviewer comment and add two phrases in the conclusion to tone it down.

Reviewer 2 Report

Authors described that they conducted expression profiles of the unfolded protein response (UPR) to clarify associations between the unfolded protein response (UPR) status and rhabdomyosarcoma (RMS) subtypes in 192 RMS cases. In this study they employed immunohistochemistry in human RMS to evaluate the expression of key UPR proteins in the four main RMS subtypes including alveolar (ARMS), embryonal (ERMS), pleomorphic (PRMS) and sclerosing/spindle cell (SRMS) RMS. In the results, the expression of BiP (GRP78), spliced XBP1 (sXBP1, nuclear XBP1), and IRE1α, were significantly associated with RMS in the four main RMS subtypes. Authors concluded the subtype and stage-specific dependency on the UPR machinery in RMS might have possibilities for the development of novel targeted therapeutic strategies and identification of specific tumor markers in RMS. I thought this article provides novel clinical and research information into the treatment of RMS patients. However, there is a minor concern.

Minor)

1)Authors should provide information of Kaplan-Meier curves regarding prognosis analyses in RMS.

Author Response

Authors described that they conducted expression profiles of the unfolded protein response (UPR) to clarify associations between the unfolded protein response (UPR) status and rhabdomyosarcoma (RMS) subtypes in 192 RMS cases. In this study they employed immunohistochemistry in human RMS to evaluate the expression of key UPR proteins in the four main RMS subtypes including alveolar (ARMS), embryonal (ERMS), pleomorphic (PRMS) and sclerosing/spindle cell (SRMS) RMS. In the results, the expression of BiP (GRP78), spliced XBP1 (sXBP1, nuclear XBP1), and IRE1α, were significantly associated with RMS in the four main RMS subtypes. Authors concluded the subtype and stage-specific dependency on the UPR machinery in RMS might have possibilities for the development of novel targeted therapeutic strategies and identification of specific tumor markers in RMS. I thought this article provides novel clinical and research information into the treatment of RMS patients. However, there is a minor concern.

Answer: All authors really appreciate the respected reviewer positive feedback on our manuscript which really encouraged and inspired all authors.

Minor)

Question 1: Authors should provide information of Kaplan-Meier curves regarding prognosis analyses in RMS.

Answer: We appreciate the respected reviewer comments. We actually planned to do it in the first version of our paper. As we used commercial “Tissue Microarray”, unfortunately, the data was not available from the company and we did not have any data to prepare Kaplan-Meier curves. We asked the company again and they were not able to provide us any further data more than the ones we have used in our manuscript.